

# Electroencephalographic power spectrum patterns related to the intelligence of children with learning disorders

Benito Javier Martínez-Briones[1], Thalía Fernández[2] and Juan Silva-Pereyra[1]

[1] Facultad de Estudios Superiores Iztacala, Universidad Nacional Autónoma de México, Tlalnepantla, México, Mexico
[2] Departamento de Neurobiología Conductual y Cognitiva, Instituto de Neurobiología, Universidad Nacional Autónoma de México, Querétaro, Mexico

## ABSTRACT

Children with learning disorders (LD) perform below average in tests of academic abilities and intelligence. These children also have a significantly abnormal resting-state electroencephalogram (EEG) compared to children with typical development (TD), *i.e.*, an excess of slow brain oscillations such as delta and theta that may be markers of inefficient cognitive processing. We aimed to explore the relationship between the performance in an intelligence test and the resting-state EEG power spectrum of children with LD. Ninety-one children with LD and 45 control children with TD were evaluated with the Wechsler Intelligence Scale for Children 4th Edition (WISC-IV) test of intelligence and a 19-channel EEG during an eyes-closed resting-state condition. The EEG dimensionality was reduced with a principal component analysis that yielded several components representing EEG bands with functional meaning. The first seven EEG components and the intelligence values were analyzed with multiple linear regression and a between-group discriminant analysis. The EEG power spectrum was significantly related to children's intelligence, predicting 13.1% of the IQ variance. Generalized delta and theta power were inversely related to IQ, whereas frontoparietal gamma activity was directly related. The intelligence test and the resting state EEG had a combined 82.4% success rate to discriminate between children with TD and those with LDs.

## INTRODUCTION

A child is diagnosed with a learning disorder (LD) if the scores on standardized academic tests of reading, writing, and/or mathematics are significantly below the expected for their age, with learning difficulties that substantially interfere with school performance or activities of daily living. Such difficulties cannot be better explained by intellectual disabilities, inadequate school instruction, or sensory/neurological impairments (*American Psychiatric Association, 2022*). With a prevalence of 5–20% in children and adolescents, LD is more common than ADHD or other neurodevelopmental disorders (*Altarac & Saroha, 2007*; *Lagae, 2008*). A reading disorder (such as dyslexia) is the most prevalent LD subtype,

Corresponding author
Juan Silva-Pereyra,
jsilvapereyra@gmail.com

but it appears in combination with other disorders (writing or mathematics) in up to 80% of LD cases (*American Psychiatric Association, 2022*; *Willcutt et al., 2013*). Such combined types of LDs were formerly known as LDs not otherwise specified (*American Psychiatric Association, 2000*).

The often-found combination of learning disorders is accompanied by a heterogeneous frame of cognitive impairments, with several cognitive processes recognized as affected, such as verbal comprehension, processing speed, and working memory (*Johnson et al., 2010*; *Semrud-Clikeman, 2005*; *Willcutt et al., 2013*). An impaired working memory has been singled out as the underlying factor behind individuals with LD's defective acquisition of academic skills (*Peng et al., 2018*; *Schuchardt, Maehler & Hasselhorn, 2008*; *Swanson, 2020*). However, a more general factor might be in place after examining the working memory hypothesis under the light of intelligence research.

## The role of intelligence in academic performance

Cognitive processes are often assessed by and in conjunction with a battery of intelligence tests, which mainly yield an IQ score as a standardized and age-corrected estimate of general cognitive ability ($g$). The g factor is an individual's proclivity to solve any set of abstract problems with a similar level of performance (*Deary, 2020*). The compiled problems of an IQ test also represent a person's ability to benefit from instruction and the amount of training needed to reach a certain level of competence (*Johnson, Deary & Carothers, 2009*). A higher IQ suggests a greater capacity for abstract reasoning and applying existing knowledge to solve problems.

The diagnosis of LD involves excluding intellectual disability, *i.e.,* an IQ below 70. Nonetheless, when the fields of education and intelligence blend, a consistent finding is that IQ scores are highly related to performance in tests of academic abilities. Intelligence is indeed the most important predictor of school achievement, with values of 58–69% of the explained variance of academic performance (*MacCann et al., 2019*; *Mammadov, 2022*). Moreover, across all academic domains and from early childhood, $g$ has been found to have a stronger relationship to academic skills than specific cognitive processes, with $g$'s explained variance higher than the accounted by the individual processes combined (*Zaboski, Kranzler & Gage, 2018*). Thus, it has been suggested that in most children at the lower ends of mathematics and reading competencies, poor academic performance could mainly result from a lower-than-average intelligence, though not as low into the cutoff point for intellectual disability, rather than being due to specific learning problems. Accordingly, most LDs would be lower-than-average instances from a normal distribution of academic and cognitive performance (*Haier, Colom & Hunt, 2023*; *Shaywitz et al., 1992*; *Wai & Bailey, 2021*). This notion would be in line with the IQ test's initial objective, originally developed by Alfred Binet as a tool to identify children with learning disabilities and tailor interventions to their specific needs, highlighting its foundational role in advancing educational equity (*Wai & Bailey, 2021*).

Moreover, research has shown that children with LDs often face challenges with self-esteem (an aspect of self-perception, also known as self-concept), largely due to academic struggles and social comparisons (*Baumeister et al., 2003*; *McArthur et al., 2020*; *Smith &*

*Nagle, 1995*). Such findings highlight the importance of framing LD-related research in ways that support children and guide evidence-based interventions. While IQ is a valuable tool for identifying cognitive strengths and difficulties, it reflects only one aspect of a child's potential, underscoring the need for assessments and interventions that address both cognitive and emotional well-being (*Eroğlu et al., 2022*; *Martínez-Briones et al., 2023*).

## Mapping intelligence through brain imaging

Regarding the neuroscience of intelligence, IQ has been related to activation levels and measures of connectivity in brain regions from three main networks: the fronto-parietal network (FPN), the default mode network (DMN), and the attention (dorsal and ventral) network. In particular, the FPN and DMN of more intelligent individuals during cognitive tasks appear in an interplay of activation of the former coupled with deactivation of the latter. This has been regarded as an effective activation of task-related brain regions while the irrelevant ones get suppressed (*Hilger et al., 2022*). Indeed, the neural efficiency hypothesis states that more intelligent individuals would require less overall brain energy, such that less activation would imply more efficient neural functioning (*Neubauer & Fink, 2009*).

Moreover, network-based analyses have identified structural-functional similarities between baseline resting states and the neural activity underlying the performance of different cognitive tasks (*Kraus et al., 2021*), supporting a notion of the resting state as a type of task general preparation state (in contrast to task-specific states); with less network reorganization across-states being found in the more intelligent subjects. Such network structure may allow high-IQ individuals to switch faster and more efficiently between different task specific states according to varying cognitive demands (*Thiele et al., 2022*). Thus, the brains of intelligent individuals could be more efficient not only by spending less overall energy to process certain levels of complexity from the environment, but also by responding to prompts with less reconfiguration between neural states.

Lastly, it should be noted that the dynamics of the FPN, the DMN and the attention network both during resting and task-related conditions have been found to be significant predictors of intelligence test scores (*Hilger et al., 2022*). Such finding is also supported by the electroencephalogram (EEG) literature summarized below that has correlated the EEG both during rest and task related conditions with intelligence, specific cognitive processes and academic performance.

## EEG as a tool to study resting and task-related conditions

The EEG allows to identify electrophysiological correlates of cognitive functioning and specific power-spectrum patterns in both healthy and clinical populations, such as children with LDs (*Cainelli et al., 2023*; *Roca-Stappung et al., 2017*). Event-related potential (ERP) studies in healthy adults have found that the amplitudes of neural responses to verbal and visuospatial problems are negatively correlated with cognitive measures of verbal and visuospatial reasoning, respectively, consistent with the brain efficiency hypothesis (*Neubauer et al., 2005*). Then, more established findings have identified attributes of ERP components such as P3 latency and the amplitude of the mismatch negativity

(MMN), indices of mental speed and automatic discrimination processes, respectively, to be negatively correlated with intelligence (*Hilger et al., 2022*).

In resting or task-related conditions, an EEG power-spectrum analysis identifies the neurophysiological activity and decomposes it into different frequencies. These frequencies, grouped into various EEG bands, have been consistently linked to different neural states and cognitive processes. For instance, the slowest EEG band of delta (0.5–4 Hz) is normally suppressed during rest in healthy and wakeful individuals, compared to several abnormal conditions. A heightened delta at rest would indicate abnormal states of reduced alertness and a defective ability to detect salient stimuli (*Knyazev, 2012*; *Lian et al., 2023*), but a healthy task related increase of delta indicates a proper filtering of sensory stimuli that could interfere with concentration (*Harmony, 2013*). The theta (4–8 Hz) band is suppressed during the resting state of normal individuals, compared to several abnormal conditions. During tasks, it is increasingly recruited according to the resources needed to properly perform a cognitive task, but is even more pronounced in less apt or mature individuals, including situations that require a higher short-term memory load and when focusing involves more effort (*Eschmann, Bader & Mecklinger, 2018*; *Gevins et al., 1997*).

Delving into faster EEG patterns, a posterior alpha (8–13 Hz) power increase appears at rest (with eyes closed while awake), a sign of healthy vigilant relaxation and disengagement from external stimuli, which can be suppressed in several abnormal conditions. Such alpha power increase has been found to be positively correlated with intelligence tests in healthy adults (*Doppelmayr et al., 2002*). Besides, a normal suppression of alpha during cognitive tasks, known as an event-related desynchronization (ERD) (*Pfurtscheller & Klimesch, 1992*), has been found to be negatively associated with intelligence during moderately difficult cognitive tasks (*Hilger et al., 2022*). Then, when alpha power is divided into lower (8–10 Hz) and upper-alpha (10–12 Hz), a typical increase in the upper part of alpha during cognitive tasks has been related to inhibitory control in the active withholding of responses (*Klimesch, Sauseng & Hanslmayr, 2007*).

Moving on to beta (13–30 Hz) power, it is related to a top-down activation of states of readiness for action and manipulation of task-related contents (*Spitzer & Haegens, 2017*), with increases in upper-beta (20–30 Hz) being linked to the preparation of motor responses (*Schapkin et al., 2020*). Lastly, gamma (30–100 Hz) power underlies cognitive functions such as perception and memory formation, which are implicated in the binding together of different sensory inputs and recollections (*Başar, 2013*; *Honkanen et al., 2015*). The increased frontoparietal gamma power during rest, identified as part of the dorsal attention network, was found to correlate with measures of arousal and was suppressed in subjects with ADHD compared to controls (*Barry et al., 2010*; *Tombor et al., 2019*).

Thus, for both healthy and abnormal populations, specific EEG patterns have been found related to cognitive processes and intelligence. In children with LDs, as summarized below, EEG patterns have been related to academic performance, but a direct relationship between intelligence and the EEG of children with LDs has not been explored.

## Previous EEG studies on learning disorders

As a preliminary note, children with LD often show a resting state with abnormal characteristics, such as elevated delta and theta activity, EEG patterns which would be indicative of general neurophysiological dysregulations that overlap with other developmental or neurological conditions, rather than being specific to LD alone. Additionally, transient physiological factors, such as metabolic variations or inflammation, can temporarily influence such EEG patterns (*Lin, 2005*). However, our study focuses on stable, group-level psychophysiological features rather than short-term fluctuations.

In particular, and compared to children with typical development, the EEG resting state of children with LDs often shows an excess of theta power (*Fernández et al., 2002*; *Jäncke & Alahmadi, 2016*), coupled with excesses of delta (*Arns et al., 2007*), beta (*Jäncke et al., 2019*), although a suppressed alpha activity (*Fernández et al., 2002*; *Fonseca et al., 2006*; *Harmony et al., 1990*). A recent meta-analysis has confirmed the presence of generalized excesses of delta and theta, and suppressed alpha activity (*Cainelli et al., 2023*). During a child's development into an adult, delta and theta power are expected to diminish, while alpha and beta power tend to increase. Then, given that the EEG of children with LDs appears abnormally slower compared to that of age-matched controls with typical development (*i.e.,* an LD's EEG is frequently similar to what is expected in younger control children), LDs have been reframed *via* the EEG maturation hypothesis as suffering from a lagged brain functional development that affects the ability to keep up at school (*Chabot et al., 2001*; *Fonseca et al., 2006*; *Guhan Seshadri et al., 2023*).

Two works that distinguished children with LDs by levels of academic impairment found the worst performers with excesses of delta, theta, and beta (*Bosch-Bayard et al., 2018*; *Roca-Stappung et al., 2017*), and also a suppressed gamma (*Roca-Stappung et al., 2017*); being notable that the children identified with more severe LDs had a lower IQ in comparison (*Cárdenas et al., 2021*; *Fernández et al., 2014*; *Harmony et al., 1990*; *Roca-Stappung et al., 2017*). Lastly, the resting state of children with LDs has also been associated with performance in academic tests. Academic ability was found to be positively correlated with alpha, and negatively correlated with delta and theta (*Harmony et al., 1990*), and another work that studied reading tasks found performance positively correlated with posterior alpha (*Babiloni et al., 2012*).

In summary, while no direct relationship between intelligence and the EEG of LDs has been studied, academic ability is highly related to intelligence, and previous studies of resting state EEG in children with LDs have found that academic performance is negatively related to delta, theta, and beta power, and positively related to alpha and gamma power. This way we could draw indirect predictions between intelligence and the functioning of brains with varying performance in academic evaluations.

Thus, unlike earlier work that primarily focused on academic performance, our study uniquely aimed to explore the association between the resting state EEG power spectrum and the intelligence of children with LDs *via* regression analysis. In line with the reported academic performance findings, our main hypothesis was that the IQ of children with LD would be negatively related to delta, theta, and beta and positively related to alpha and gamma power. As a secondary objective, we explored if any of the intelligence and EEG

variables can adequately discriminate between an LD or a control child *via* a discriminant analysis, to clarify the relevance of intelligence and EEG measures in studying learning disorders.

## MATERIALS & METHODS

### Participants

This study's protocol complies with the Ethical Principles for Medical Research Involving Human Subjects established by the Declaration of Helsinki (*World Medical Association Declaration of Helsinki, 2013*). All the children as well as their parents signed informed consent forms that were approved by the Ethical Committee of the Instituto de Neurobiología (INEU/SA/CB/146) of the Universidad Nacional Autónoma de México (UNAM). The parents agreed to the use of the information, which was further re-coded (anonymized) to protect the privacy of the subjects, for research and academic dissemination purposes.

A total of 136 right-handed children (59 girls) between the ages of 8 and 11 participated in this study. These participants were recruited from both public and private elementary schools in Mexico. To be included in the study, all children needed to meet the following three criteria:

1. A normal neurological and psychiatric assessment (with the exception of LD-specific diagnostic criteria detailed below), no language impairments, and no uncorrected visual or auditory deficits (children with vision issues were required to wear corrective glasses).
2. An intelligence quotient (IQ) score above 70, as determined by the Wechsler Intelligence Scale for Children 4th Edition (WISC-IV), standardized for the Mexican population (*Fina, Sánchez-Escobedo & Hollingworth, 2012*; *Wechsler, 2010*), to exclude intellectual disabilities
3. No severe socioeconomic disadvantages, defined as having a mother (or primary guardian) with at least an elementary school education and a per capita income above 50% of the minimum wage.

To minimize potential confounds related to transient physiological fluctuations, all participants were in normal health at the time of testing, and any suspected cases of flu-like illness were rescheduled. Additionally, a general interview with parents regarding illness history was conducted to screen for acute conditions that could temporarily impact cognitive function. None of the participants were taking medication for neurological or psychiatric conditions at the time of testing.

Within the sample, 91 children (34 girls) were identified as having a learning disorder based on the following three criteria: (a) low academic performance reported by both teachers and parents; (b) scores below the 10th percentile on reading, writing, or mathematics subscales of the Neuropsychological Scale for Children 2nd Edition (ENI-2), standardized for Mexican children (*Matute et al., 2014*); and (c) a psychologist's final diagnosis of LD based on DSM-5 criteria (*American Psychiatric Association, 2022*). Although some children missed completing items on the attentional evaluation, none met DSM-5 criteria for ADHD (*American Psychiatric Association, 2022*).

**Table 1  Between-group *t*-test comparison of the main descriptive characteristics of the participants.**
The reading, writing, and mathematics variables are composite percentile scores from the ENI-2 scale.

| Test | Variable | Ctrl group n = 45 Mean (SD) | LD group n = 91 Mean (SD) | t | p |
|---|---|---|---|---|---|
| WISC-IV | Age | 9.53 (0.89) | 9.25 (0.93) | 1.68 | 0.095 |
|  | Full scale IQ | 105.73 (12.68) | 89.19 (10.07) | 8.26 | <0.0001 |
| ENI-2 | Reading | 203.29 (36.91) | 71.49 (57.43) | 14.06 | <0.0001 |
|  | Writing | 210.16 (40.90) | 92.30 (59.18) | 12.01 | <0.0001 |
|  | Mathematics | 265.73 (58.22) | 125.97 (68.67) | 11.72 | <0.0001 |

The remaining 45 children (25 girls) comprised the control group, consisting of typically developing children with good academic performance as reported by their parents and teachers. These children scored above the 36th percentile in reading, writing, and mathematics on the ENI-2 scale (*Matute et al., 2014*). Table 1 presents the key descriptive statistics for the two groups.

## Indices of the WISC-IV and ENI-2 scales

The WISC-IV comprises 15 subtests that together provide a full-scale IQ score, derived from four core indices:

- Verbal Comprehension Index (VCI): Assesses verbal abilities, including vocabulary, comprehension, and general knowledge. This index is supported by the subtests Similarities, Vocabulary, and Comprehension.
- Perceptual Reasoning Index (PCI): Evaluates nonverbal reasoning skills, such as visuospatial abilities, perceptual organization, and problem-solving. The supporting subtests for this index are Block Design, Picture Concepts, and Matrix Reasoning.
- Working Memory Index (WMI): Measures the ability to temporarily retain and manipulate information. The subtests associated with this index include Digit Span and Letter–Number Sequencing.
- Processing Speed Index (PSI): Reflects the ability to perform simple tasks quickly and efficiently. Coding and Symbol Search are the subtests used for this index.

With the ENI-2 test, a general cognitive test for children, we obtained percentile composite scores for the following academic domains:

- Reading ability: Measures a child's precision, comprehension, and speed of reading *via* several subtests, such as reading of syllables, words, nonwords, sentences, and a short story.
- Writing ability: Measures a child's precision, narrative composition, and writing speed *via* several subtests, such as writing the child's own name; dictation of syllables, words, nonwords, sentences, and a short story; and the precision to copy and recall an additional story.
- Mathematical ability: Measures a child's ability to count, compare, calculate, and use logical-mathematical reasoning *via* subtests such as counting, number reading, number dictation, number comparison, ordering, doing forward and backward number

**Table 2** Correlation matrix between five intelligence variables and three variables of academic performance.

|  |  | Reading | Writing | Mathematics |
|---|---|---|---|---|
| VCI | Pearson's r | 0.530[*] | 0.532[*] | 0.545[*] |
|  | Original *p*-value | 3.14E−11 | 2.70E−11 | 7.13E−12 |
|  | FDR adjusted *p*-value | 5.17E−11 | 4.73E−11 | 1.66E−11 |
| PRI | Pearson's r | 0.314[*] | 0.323[*] | 0.487[*] |
|  | Original *p*-value | 1.98E−04 | 1.25E−04 | 1.84E−09 |
|  | FDR adjusted *p*-value | 1.98E−04 | 1.39E−04 | 2.70E−09 |
| WMI | Pearson's r | 0.544[*] | 0.532[*] | 0.588[*] |
|  | Original *p*-value | 7.71E−12 | 2.63E−11 | 5.02E−14 |
|  | FDR adjusted *p*-value | 1.66E−11 | 4.73E−11 | 1.56E−13 |
| PSI | Pearson's r | 0.316[*] | 0.352[*] | 0.362[*] |
|  | Original *p*-value | 1.81E−04 | 2.63E−05 | 1.45E−05 |
|  | FDR adjusted *p*-value | 1.88E−04 | 3.20E−05 | 1.85E−05 |
| Full-Scale IQ | Pearson's r | 0.566[*] | 0.572[*] | 0.657[*] |
|  | Original *p*-value | 6.95E−13 | 3.40E−13 | 3.76E−18 |
|  | FDR adjusted *p*-value | 1.77E−12 | 9.51E−13 | 2.11E−17 |

**Notes.**

The correlation matrix shows pairwise relationships among the variables. The *p*-values were corrected for multiple comparisons using the Benjamini–Hochberg False Discovery Rate (FDR) method at $q = 0.05$.

[*]Significant correlations after correction appear with an asterisk.

VCI, Verbal Comprehension Index; PRI, Perceptual Reasoning Index; WMI, Working Memory Index; PSI, Processing Speed Index.

sequences, mental calculation, written calculation and solving sentences of mathematical problems.

Both scales have been previously used to study and diagnose children with LDs (*Fernández et al., 2002*; *Martínez-Briones et al., 2020*; *Roca-Stappung et al., 2017*). Since this work assumes that academic and intelligence scores are highly related, Table 2 shows a correlation matrix between both scales.

## EEG acquisition and data analysis

A 19-channel EEG (Ag/AgCl electrodes held by a cap according to the 10–20 International System; Electro-Cap International Inc., OH, USA), referenced to linked earlobes (A1A2), was recorded during a continuous 10-minute eyes-closed resting state condition, with a MEDICID IV system (Neuronic S.A.; Mexico City, Mexico) and a Track Walker v5.0 data system. In a dim-lit, air-conditioned, sound-proofed, and faradized room, the children were comfortably seated while being prompted to breathe naturally, stay relaxed, and fully awake. The amplifier's bandwidth was set between 0.5 and 50 Hz. All electrode impedances were at or below 5 kΩ, and the signal was amplified with a gain of 20,000. The EEG data, re-coded to omit the name of the subjects, was sampled every 5 ms and edited offline by an expert neurophysiologist, who manually selected artifact-free and quasi-stationary epochs without the assistance of automatic algorithms for artifact rejection. Automatic artifact rejection is more useful for high-density EEG recordings, where the visual inspection of the data is considerably difficult. However, since our recordings were based on the standard

setting of 19 channels, we opted to control the EEG editing to obtain clean enough recordings based on an expert's criteria, thus avoiding automatic procedures which are not 100% guaranteed to produce a clean signal and which may also introduce undesirable effects on the data (*Martínez-Briones et al., 2020*).

Twenty-four artifact-free segments of 2.56 s were selected from each child. This segment length is commonly used both in clinical and experimental EEG studies since it avoids non-stationarities in the EEG signal and guarantees an appropriate frequency resolution of 0.39 Hz for the analysis. The 24 segments represent an accurate spectral description of the EEG signals, with more degrees of freedom than the number of electrodes, adding to at least a minute of representative EEG activity (*Başar & Dumermuth, 1982*; *Bosch-Bayard et al., 2018*).

A diagram of the following EEG processing steps is shown in Fig. 1. The EEG data was subjected to a Fast Fourier Transform (FFT) to obtain a 0.78–50 Hz power spectrum with a frequency resolution of 0.39 Hz, and then corrected with a global scale factor (*Hernández et al., 1994*). The 127 narrow bands obtained per channel were grouped into seven broad bands: delta (0.7–3.6 Hz), theta (3.9–7.5 Hz), alpha-1 (7.8–9.8 Hz), alpha-2 (10.1–12.6 Hz), beta-1 (12.8–20.4 Hz), beta-2 (20.7–30.47 Hz), and gamma (30.85-50 Hz).

To further reduce the dimensionality of the EEG data (now seven bands by 19 channels), a principal component analysis (PCA) with the varimax rotation that generates orthogonal components, *i.e.,* independent enough to avoid collinearity in the regression analysis (*Arruda et al., 1996*; *Ferrari-Díaz et al., 2022*), was applied over the 133 variables. The Kaiser criterion (*Kaiser, 1960*) yielded 23 components that explained 87.53% of the total variance. Upon inspection, the first seven components represented functionally meaningful bands that explained 60.57% of the variance (see the PCA results in Data S1). They were distributed as follows:

Generalized (from most EEG channels) delta and theta power were better represented by Component 1 (C1). Generalized alpha-2 was represented by Component 2 (C2). Generalized alpha-1 was represented by Component 3 (C3). Generalized beta-1 was represented by Component 4 (C4). Frontocentral beta-2 was represented by Component 5 (C5). Frontal theta was represented by Component 6 (C6). Lastly, frontoparietal gamma was represented by Component 7 (C7).

The remaining sixteen components were excluded from the regression analysis due to being composed of the last 33 (out of 133) of the scattered channels of the theta, alpha, beta, or gamma bands, with none of the components being composed of more than two channels of the same frequency band, and thus providing no functional meaning to draw on. In contrast, the first seven selected components were composed of at least five channels of the same frequency band (C6), up to 29 channels of the same frequency band (C1).

To validate the appropriateness of the multiple regression analysis, we tested and met the assumptions of normality, homoscedasticity and no-multicollinearity. To this effect, a normal PP-plot (Fig. 2, left panel) shows that the regression standardized residual data greatly follows the normality line. Then, a scatterplot to check for homoscedasticity (Fig. 2, right panel) shows the lack of a clear pattern or skew, consistent with what homoscedastic data would look like. Finally, Table 3 shows that the variance inflation factor (VIF) values

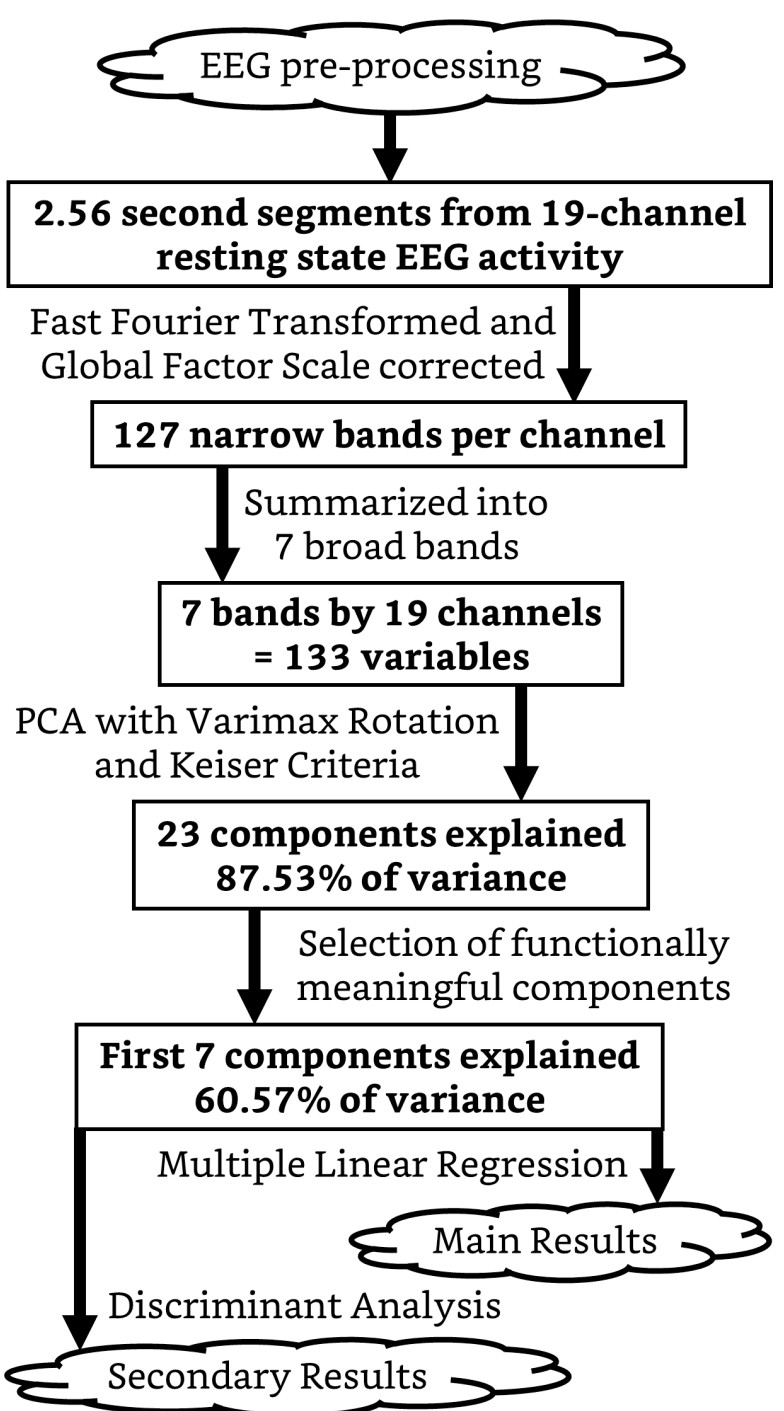

**Figure 1 Diagram of the EEG processing steps.** Boxes represent the data structure in a given state. Arrows represent a type of processing applied to the data. PCA, principal component analysis.

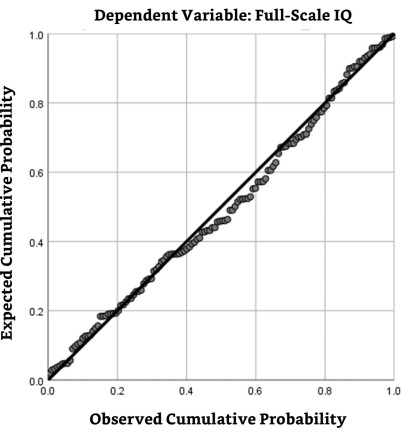

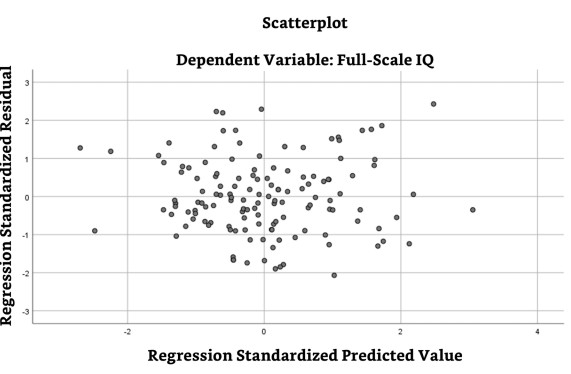

**Figure 2 Normal P-P plot and homoscedasticity scatterplot.** The normal P-P plot of left panel shows that the observed cumulative probability greatly follows the expected cumulative probability, a pattern consistent with an acceptable level of normality of the data. The scatterplot of the right panel shows a patternless spread of the data, consistent with what homoscedastic data would look like.

**Table 3 Coefficients of the seven components for the regression model 1.**

| Component | β | 95% CI | | p | VIF |
|---|---|---|---|---|---|
| | | LL | UL | | |
| C1 | −3.66 | −5.85 | −1.47 | 0.001 | 1.00 |
| C2 | −0.97 | −3.16 | 1.23 | 0.386 | 1.00 |
| C3 | −1.10 | −3.30 | 1.09 | 0.321 | 1.00 |
| C4 | −0.92 | −3.12 | 1.27 | 0.408 | 1.00 |
| C5 | −0.47 | −2.66 | 1.73 | 0.675 | 1.00 |
| C6 | −0.71 | −2.90 | 1.48 | 0.523 | 1.00 |
| C7 | 2.55 | 0.36 | 4.75 | 0.023 | 1.00 |

**Notes.**
CI, confidence interval; LL, lower limit; UL, upper limit; VIF, variance inflation factor.

to test for multicollinearity were at 1.00, with values below 5.00 indicating minimal concern for multicollinearity.

A note on between-group sex differences: Since girls comprised 37% of the LD group and 56% of the control group, a chi-square test revealed statistically significant sex differences ($X^2 = 4.06$, $p = 0.044$). Such was an expected result, given that LDs are consistently found to be more prevalent in boys than girls (*American Psychiatric Association, 2000*; *American Psychiatric Association, 2022*). Thus, in a stepwise multiple linear regression analysis, the first seven EEG components were included as predictors to the Full-Scale IQ, and in the next step, the sex variable was added as a predictor.

A discriminant analysis was performed to further explore if the intelligence scores and the EEG components effectively distinguish between children with LDs and controls. In this analysis, next to the seven EEG components we added the four intelligence indices of the WISC-IV test: Verbal Comprehension (VC), Perceptual Reasoning (PR), Working

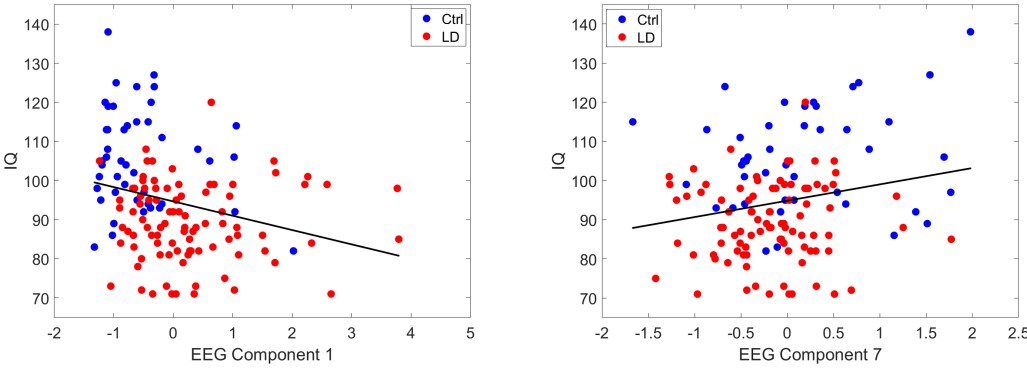

**Figure 3  Partial regression plots of the EEG components significantly related to IQ.** The EEG component 1 (C1) represents generalized delta and theta activity, while the EEG component 7 (C7) represents frontoparietal gamma activity. C1 had a significant inverse relationship to IQ while C7 was directly related. The different colors of the data points are only to highlight the contrast between the groups: subjects from the control (Ctrl) group appear in blue, while subjects from the group with learning disorders (LD) appear in red.

Memory (WM), and Processing Speed (PS). One limitation of discriminant analysis is its tendency to overfit when the number of predictors is large relative to the sample size, which may result in overly optimistic classification accuracy. To address this, we employed jackknife classification (leave-one-out cross-validation), a robust cross-validation method that systematically tests each observation by excluding it from the training set and using the remaining data to build the model. This approach mitigates overfitting by ensuring that the classification performance is validated on unseen data for each iteration, providing a more reliable estimate of the model's generalizability (*Kuligowski, Pérez-Guaita & Quintás, 2016*).

Statistical analyses were conducted using the IBM SPSS Statistics 26.0 software (SPSS Inc., Armonk, NY, USA) for Windows. Figure 3 was composed with MATLAB (The MathWorks Inc., Natick, MA, USA) version 9.13.0.2105380 (R2022b).

## RESULTS

The four WISC-IV indices and the Full-Scale IQ were strongly correlated to the percentile scores of reading, writing, and mathematics (see Table 2). Correlation-based effect sizes (*Maher, Markey & Ebert-May, 2013*) yielded medium (perceptual reasoning and processing speed $r$-values were between .3 and .5) to large (verbal comprehension, working memory, and full-scale IQ $r$-values were above .5) effect sizes between the intelligence indices and the academic performance scores. This was a finding expected to be replicated in this sample of children, given the evidence of a strongly established relationship between intelligence and academic ability.

To assess the relationship between the EEG components and the Full-Scale IQ, two regression models, with and without the inclusion of sex, indeed showed the same direction of results. Model 1 (without sex) yielded a significant regression equation ($F(7, 128) = 2.75$, $p = 0.011$), with an $r^2$ of 0.131. Significant individual EEG components identified were

**Table 4  Classification results table.**

| | | Predicted group membership (%) | | |
| --- | --- | --- | --- | --- |
| | **Main Groups** | **Ctrl** | **LD** | **Total** |
| Original | Ctrl | 71.1 | 28.9 | 100 |
| | LD | 4.4 | 95.6 | 100 |
| Cross-validated | Ctrl | 60 | 40 | 100 |
| | LD | 6.6 | 93.4 | 100 |

Notes.
Original grouped cases correctly classified = 87.5%; cross-validated grouped cases correctly classified = 82.4%, *via* the jack-knife classification method.

C1 (generalized delta and theta) with an inverse relationship ($\beta = -3.66$, 95% CI [$-5.85$ to $-1.47$], $p = 0.001$) to IQ, and C7 (frontoparietal gamma) with a direct relationship ($\beta = 2.55$, 95% CI [0.36–4.75], $p = 0.023$) to IQ (see Table 3).

Model 2 (with sex) was similar to Model 1, showing a significant regression equation ($F(8, 127) = 2.39$, $p = 0.02$), with an $r^2$ of 0.131. There was no significant contribution of sex ($\beta = 0.47$, 95% CI [$-5.08$–4.15], $p = 0.84$). Moreover, the only significant individual EEG components identified were also C1 with an inverse relationship ($\beta = -3.70$, 95% CI [$-5.93$ to $-1.46$], $p = 0.001$) to IQ, and C7 with a direct relationship ($\beta = 2.59$, 95% CI [.36–4.83], $p = 0.024$) to IQ (see also Fig. 3). It should be noted that sex was included as a predictor in this regression model due to its established association with LD prevalence, with boys being more frequently diagnosed than girls. However, this finding of sex being no significant predictor indicates that sex does not necessarily affect cognitive or neurophysiological measures of pre-pubertal children such as those examined in this study. Rather, sex differences in the EEG could mainly start to appear during adolescence (*Campbell et al., 2005*). Thus, the observed patterns in EEG and intelligence were likely independent of sex-based differences.

A discriminant analysis was conducted to predict whether a child had an LD or not. Predictor variables were the WISC-IV indices and the first seven EEG components. Significant mean differences between groups were observed for the following predictors: the four WISC-IV indices (VC, PR, WM, PS; $p$-values < 0.0001) and the first and seventh EEG components ($p$-values: <0.0001, and 0.002). The discriminate function revealed a significant association between groups and the predictors mentioned above, accounting for 47.6% of between-group variability; however, a closer analysis of the structure matrix revealed six significant predictors, namely VC (0.68), WM (0.60), C1 ($-0.46$), PS (0.39), PR (0.38), and C7 (0.28); the other five EEG components were poor predictors. The cross-validated classification showed that overall (see Table 4), 82.4% of cases were correctly classified (control group cross-classification = 60%; LD cross-classification = 93.4%). Thus, performance in the WISC-4 test, coupled with EEG bands such as delta, theta, and gamma, was adequate for identifying children with LD.

## DISCUSSION

Our research has shed light on the relationship between intelligence and the EEG of children with learning disorders. Unlike earlier work that primarily focused on academic performance, our study uniquely aimed to explore the association between the resting state EEG power spectrum and the intelligence of children with LDs *via* regression analysis. In line with the reported academic performance findings, our main hypothesis was that the IQ of children with LD would be negatively related to delta, theta, and beta and positively related to alpha and gamma power. The main finding of this work is that the resting state EEG is a significant predictor of intelligence, accounting for 13.1% of the IQ variance among school-age children. This prediction is primarily influenced by the contributions of some EEG bands, such as generalized delta and theta, which show an inverse relationship with IQ, and frontoparietal gamma power, which is directly related. Additionally, a secondary result of a discriminant analysis revealed that children with LDs exhibit generalized excesses of both delta and theta power in the resting-state EEG and a suppressed frontoparietal gamma power when compared to controls.

The bands in the EEG power spectrum were initially defined based on brain rhythms. Different rhythms have distinct origins, functional meanings, and behavior according to a given condition (*Başar, 2013*; *Fernández et al., 2000*; *Knyazev, 2012*; *Mitchell et al., 2008*). The most situation-sensitive band is alpha, appearing with power increases at rest with eyes closed in posterior sites, and it increases more throughout development; alpha desynchronizes with eyes open, and such level of desynchronization changes according to the difficulty of a given cognitive task (*Pfurtscheller, Neuper & Berger, 1994*). Considering the findings of greater alpha power at rest correlated with intelligence (*Doppelmayr et al., 2002*) and the children with LDs being often found with reduced alpha compared to controls, we expected to find differences between groups (*via* the discriminant analysis) and for alpha to be directly correlated with intelligence. This and an opposite tendency for the beta band (*Bosch-Bayard et al., 2018*), expected to relate inversely to intelligence, were the only nonsignificant results that differed from our main hypothesis. It should be noted, however, that alpha power has mixed support in its relationship with intelligence (*Hilger et al., 2022*), and our PCA analysis generalized alpha through most channels, possibly missing an effect at posterior sites. The same could apply to the beta band in relationship to the PCA, beta being somewhat generalized in this work but whose effect may appear more restricted in central channels. More limitations of this work are summarized below, including the notion that a PCA would improve its precision with higher density arrays and by analyzing the EEG at the sources (*Fernández et al., 2002*; *Martínez-Briones et al., 2020*).

Moving on to interpret the slower bands (delta and theta) found statistically significant in this work, academic ability indeed has been previously found negatively correlated with delta and theta (*Harmony et al., 1990*), and by separating children with LDs by levels of academic impairment, the worst performers do show excesses of delta and theta (*Bosch-Bayard et al., 2018*; *Roca-Stappung et al., 2017*). On the one hand, delta is normally suppressed at rest in healthy individuals, but its increase could indicate reduced alertness

linked to diminished cognitive readiness, during which the individual is more likely to make mistakes if presented with a cognitive task (*Knyazev, 2012*; *Lian et al., 2023*); in line with this, *Fernández et al. (1998)* reported that a higher frontal delta activity prior to stimulus presentation predicted incorrect responses in a continuous performance task in children. On the other hand, the theta band is normally reduced in healthy individuals compared to abnormal populations, and its increase is associated with more effortful inefficiencies in less apt or more immature individuals (*Eschmann, Bader & Mecklinger, 2018*; *Martínez-Briones et al., 2020*). Nonetheless, the increases in delta and theta without a reduced alpha power found in this work do not fully support nor exclude the EEG maturation hypothesis of lagged neurofunctional development in learning disorders (*Chabot et al., 2001*; *Fonseca et al., 2006*; *Guhan Seshadri et al., 2023*). Considering the neural efficiency hypothesis, which states that more intelligent individuals exhibit a reduced brain activation reflecting greater efficiency, and adapting it to children with learning disorders (LD), this suggests that abnormal EEG patterns, such as excessive delta and theta activity, may reflect inefficient neural functioning that contributes to their cognitive challenges. Thus, this implication for children with LD frames the observed EEG abnormalities as potential markers of impaired cognitive efficiency, rather than purely maturational delays.

Then, in support of our findings of gamma power being decreased in LDs while directly related to intelligence, gamma has indeed been reported suppressed in worst academic performers compared to better performers (*Roca-Stappung et al., 2017*). Also, frontoparietal gamma has been found to be inversely correlated with measures of arousal *via* a possible link to the dorsal attentional network (*Barry et al., 2010*; *Tombor et al., 2019*). Then, the direct relationship found between frontoparietal gamma activity and IQ aligns with views regarding the role of attentional networks in higher-order cognitive processes, given that gamma power is thought to underlie neural processes associated with sustained attention, working memory, and information integration (*Barry et al., 2010*; *Başar, 2013*). However, while a suppressed frontoparietal gamma might point to impaired attentional integrity, its functional role should be clarified in future studies since gamma has been barely studied in populations of children with LDs, given that most of the literature on LDs does not go beyond the beta band.

The intelligence indices and the resting state EEG had a combined 82.4% success rate to discern between control children and those with LDs. The significant discriminating factors, ranked in order of importance as classifiers, were Verbal Comprehension, Working Memory, EEG Component 1 (generalized delta and theta), Processing Speed, Perceptual Reasoning, and EEG Component 7 (frontoparietal gamma). This result would add to the critique of the working memory hypothesis as the cognitive factor underlying LDs (*Martínez-Briones, Fernández & Silva-Pereyra, 2023*). Working memory does contribute to LDs, but not as much as verbal comprehension or all the cognitive indices combined that are factored by g (*Zaboski, Kranzler & Gage, 2018*). Then, the finding in this work of all the intelligence indices strongly correlated to the academic scores of reading, writing, and mathematics might add up to consider the positive manifold tendency towards g as a primary driver of LDs (*Haier, Colom & Hunt, 2023*).

The above would suggest that the DSM-5 and clinical practice could consider intelligence as more than an exclusion criterion in diagnosing LDs. In a normal distribution of intelligence, LDs often have lower-than-average IQs. While not intellectually disabled, they still do require extra attention and resources to increase their opportunities *via* special education programs and improvements in social policy (*Wai & Bailey, 2021*), *e.g.*, the recently established "national dyslexia awareness month" (*Shaywitz & Shaywitz, 2020*). Also, an intelligence test and the resting state EEG might provide markers that are good enough to identify children with LDs. Both are relatively cheap evaluation instruments with the potential to contribute more to the characterization, diagnosis, and treatment of children with LDs.

Lastly, we acknowledge three main limitations of this work: (1) a surface 19-channel EEG accentuates the low spatial resolution of the EEG that would be improved with higher density arrays. Since the relatively low density of electrodes does not capture fine-grained neural activity or localize the specific brain regions generating the observed signals, this results in a rough interpretation of EEG components, mainly by not distinguishing activity from adjacent or overlapping cortical areas. (2) Such drawback in special resolution would also be improved with an analysis of the sources that also overcome other problems (*e.g.*, signal leakage due to volume conduction *Biscay, Bosch-Bayard & Pascual-Marqui, 2018*) of the surface EEG used in this work. A source analysis would indeed yield a basic identification of the brain regions involved in the intelligence of these children (*Fernández et al., 2002*; *Pascual-Marqui, 2007*). Thus, future studies employing high-density EEG arrays or source localization techniques could address these limitations and provide a more detailed understanding of the neural dynamics associated with intelligence and LDs. (3) Lastly, given that the neural correlates of intelligence are organized as brain regions linked into functional networks such as the frontoparietal and the default mode networks (*Ramchandran, Zeien & Andreasen, 2019*), an effective connectivity analysis at the sources that takes advantage of the high temporal resolution of the EEG could yield, also with better spatial resolution (*Bosch-Bayard et al., 2022*), more subtle types of causal relationships between neural networks and the intelligence of children with learning disorders.

## CONCLUSIONS

Previous studies have consistently reported elevated delta and theta power and reduced alpha activity in children with LD during resting-state EEG. These findings align with our results, which extend the literature by examining the relationship between these EEG patterns and intelligence scores. Our findings corroborate and build upon the existing evidence by demonstrating that generalized delta and theta power negatively correlate with IQ, while frontoparietal gamma activity shows a positive relation.

Specifically, the resting-state EEG predicted 13.1% of the intelligence variance in school-age children. The following EEG patterns were related to the children's intelligence: Generalized delta and theta predicted a lower IQ, bands whose functional meaning may indicate impaired alertness coupled with more inefficient cognitive processing in children

with LD. Frontoparietal gamma power predicted a higher IQ, a band whose functional meaning may indicate an adequate state of arousal.

Additionally, an intelligence test and the resting state EEG were adequate to discriminate, with an 82.4% success rate, between children with typical development and LD, thus yielding such evaluations relevant enough for studying and diagnosing learning disorders.

Regarding future directions of this type of research, ventures should be directed towards improving the limitations of this work, *i.e.,* to improve spatial accuracy with higher density arrays and *via* analyses at the sources instead of the surface EEG; also by studying the EEG connectivity between the network-based regions of interest underlying intelligence. Such efforts should as well be directed towards distinguishing between different subgroups of specific learning disorders, *i.e.,* dyslexia, dysgraphia, and dyscalculia.

## ACKNOWLEDGEMENTS

The authors are grateful for the cooperation of the children and parents who participated in this study. Also, the technical assistance of Héctor Belmont is deeply appreciated. The authors declare that aspects of idea generation/exploration and language improvement (such as synthesis, paraphrasing, translation, and style correction in English), have been conducted with the assistance of the GPT-4.0 tool of generative artificial intelligence. All modifications were carefully reviewed to ensure alignment with the intended message and adherence to academic and professional standards.

### Funding

This work was supported by the Universidad Nacional Autónoma de México (UNAM) Postdoctoral Program (POSDOC). The funders had no role in study design, data collection and analysis, decision to publish, or preparation of the manuscript.

### Grant Disclosures

The following grant information was disclosed by the authors:
Universidad Nacional Autónoma de México (UNAM) Postdoctoral Program (POSDOC).

### Competing Interests

The authors declare there are no competing interests.

### Author Contributions

- Benito Javier Martínez-Briones conceived and designed the experiments, performed the experiments, analyzed the data, prepared figures and/or tables, authored or reviewed drafts of the article, and approved the final draft.
- Thalía Fernández conceived and designed the experiments, performed the experiments, authored or reviewed drafts of the article, and approved the final draft.
- Juan Silva-Pereyra analyzed the data, authored or reviewed drafts of the article, and approved the final draft.

## Human Ethics

The following information was supplied relating to ethical approvals (i.e., approving body and any reference numbers):

All participants and their parents signed informed consent forms that were approved by the Ethical Committee of the Instituto de Neurobiología (INEU/SA/CB/146) of the Universidad Nacional Autónoma de México (UNAM). Approval number INEU/SA/CB/146.

## Data Availability

The data is available in the Supplemental File and at Mendeley: Martínez-Briones, Benito Javier (2024), "EEG power spectrum and Intelligence of Children with Learning Disorders", Mendeley Data, V1, doi: 10.17632/t6xw4sdfdc.1.

## Supplemental Information

Supplemental information for this article can be found online at http://dx.doi.org/10.7717/peerj.19138#supplemental-information.

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
