# Peer review of "Electroencephalographic power spectrum patterns related to the intelligence of children with learning disorders"

_PeerJ, doi:10.7717/peerj.19138_

## Round 0.1 · original submission · Major Revisions

Dear authors,

Three reviewers called for major revisions. The comments from Reviewer 3 are the most specific; many of them involve better description of statistical methods and results. Please pay careful attention to these comments in your revised MS.

Reviewer 1 ·

Basic reporting

This study effectively correlated IQ and cognitive scores with specific dysregulated EEG frequency features. The instruments used are well-established tests for this. The surface EEG observed has been shown to be present in LD in many other studies, so replicating those findings is meaningful, especially since EEG dynamics are changing dramatically with increased computer and smart device use.

That said, I do not find these EEG features to be specific predictors since they appear in many other disorders. I ask the authors to reframe that observation to accommodate this concept.

Experimental design

Excellent and straightforward design.

Validity of the findings

The findings are valid except for my comment above.

Additional comments

No further comments.

Reviewer 2 ·

Basic reporting

I have to reject the hypothesis of this article which states that IQ in LD is less than that of the Typicals.
That is true that the delta, theta in general are above in LD compared with typicals, and gamma power is low, but it does not determine the IQ. The children with LD perform less in Wisc-R tests because their visual and auditory skills are compromised, and they can not finish completing the questiona and answers on time. On the other hand, some of tham have a very high IQ.
There are many scientists who have LD, like Einstein, Edison, Leonardo da Vinci. It means the WISC-R tests are not enough to measure the IQ of these children.
Although there may be neurodevelopmental delays for the maturation of brain, especially the cortex and the left hemisphere, the interconnectedness between the two hemispheres are high, which creates a broader intelligence.
These children's self esteem is low, so I do not see any point to publish this article which will increase the discrimination and lower their self esteem. Instead they need better intervention methods to enable them to be integrated to the life.
The article is written in Mexico, where the human rights and priviliges are not a big concern, but publishing this article hurts the human values in general.

Experimental design

None

Validity of the findings

None

Additional comments

None

Reviewer 3 ·

Basic reporting

Language and Structure:

The manuscript is generally well-written, employing professional and scientific language appropriate for an academic audience. However, certain sentences could benefit from improved clarity. For example, the presentation of previous findings on EEG correlations and intelligence in the Introduction section (Lines 33-94) is somewhat dense and could be structured more effectively. Breaking down the discussion into more manageable parts would improve readability and comprehension.

Background Context:

The introduction provides an adequate review of the relevant literature regarding EEG characteristics in children with learning disabilities (LD) and the correlations with intelligence. However, a more explicit discussion of the neural efficiency hypothesis (Lines 74-91) and its specific implications for children with LD would strengthen the conceptual framing of the study. Furthermore, previous EEG studies involving children with LD (Lines 140-158) are mentioned, but a more detailed comparison to the current work would help situate this study within the existing body of research.

Figures and Tables:

Figures: Figure 2 (Lines 326-330) provides partial regression plots that illustrate the relationships between EEG components and IQ. The colors used to differentiate the groups are not clearly explained in the figure legend, which may lead to confusion. Adding a detailed description of the color coding in the legend would improve the clarity of the figure.

Tables: The correlation matrix in Table 2 (Lines 308-311) shows significant correlations between intelligence measures and academic performance. Including effect sizes for these correlations would provide a better understanding of the practical significance of these relationships, not just their statistical significance. Also, a multiple comparisons correction (false discovery rate) is advisable. P-values should also be reported.

Experimental design

Research Question and Scope:

The research question is clearly stated in the Introduction (Lines 19-30), focusing on the relationship between resting-state EEG and intelligence in children with LD. The study is well within the scope of the journal, and it addresses a relevant and meaningful gap in understanding the neurophysiological correlates of LD.

Methods and Replicability:

The Materials & Methods section (Lines 178-296) provides a comprehensive overview of the experimental procedures, including participant recruitment, EEG acquisition, and data analysis. However, some aspects of the EEG preprocessing steps, specifically the artifact rejection criteria (Lines 254-258), lack sufficient detail. Providing more explicit thresholds or examples of how artifacts were defined and removed would enhance the replicability of the study.

Statistical Analysis:

The use of Principal Component Analysis (PCA) for dimensionality reduction (Lines 267-279) is appropriate, but the rationale for selecting the first 7 components, which explain 60.57% of the variance, could be elaborated further. Why were these 7 components chosen, and not more or fewer? A more detailed justification would strengthen the robustness of the analysis.

The multiple linear regression and discriminant analysis employed (Lines 287-342) are suitable for exploring the relationships between EEG components and intelligence scores. However, the authors should provide additional clarification regarding the inclusion of sex as a covariate (Lines 320-322). Specifically, it would be helpful to explain why sex was not found to be a significant predictor and how this finding relates to existing literature on sex differences in LD prevalence.

Validity of the findings

Data Robustness and Soundness:

The findings appear to be statistically sound, with appropriate use of multiple regression and discriminant analysis techniques. However, the authors should provide more information about the assumptions checked before running these analyses. For instance, were normality and multicollinearity assumptions tested for the regression models (Lines 287-342)? Including such details would help validate the robustness of the statistical approach.

Interpretation of Results:

The interpretation of the regression results (Lines 313-324) is clear but could benefit from a deeper discussion of the practical implications of the findings. Specifically, the inverse relationship between generalized delta and theta power and IQ should be discussed in the context of what these EEG patterns imply for cognitive functioning in children with LD. Additionally, the direct relationship between frontoparietal gamma activity and IQ should be linked to existing theories on attentional networks and intelligence (Lines 328-329).

Discriminant Analysis:

The discriminant analysis results (Lines 333-342) show a high success rate (82.4%) in distinguishing between LD and control groups. While these results are promising, the authors should discuss potential limitations of this analysis, such as the risk of overfitting given the relatively small sample size. A mention (or better, an application) of cross-validation techniques used to mitigate overfitting (if applicable) would also be valuable.

Limitations and Future Directions:

The discussion section touches on some limitations (Lines 421-431), but it would benefit from a more thorough elaboration. For example, the limitation related to the use of a 19-channel EEG system (Line 421) could be expanded to discuss how this may impact spatial resolution and the interpretation of EEG components. Additionally, suggestions for future research should include more specific recommendations, such as using higher-density EEG arrays or source localization techniques to improve spatial accuracy.

Data Availability and Ethical Considerations:

The manuscript mentions that the data was collected in compliance with ethical guidelines (Lines 181-185). However, more detail on how participants' data privacy was ensured would enhance the transparency of the ethical considerations. For example, were data anonymized, and how was consent obtained?

---

## Round 0.2 · Minor Revisions

Reviewer 2 highlights that external factors, such as illness/inflammation may affect both EEG and IQ measurements. I suggest you add a note within the region the reviewer suggests discussing this in the context of your experimental design and results.

Reviewer 2 has suggested you evaluate and possibly cite a reference which looks at biofeedback manipulation of theta and delta in LD versus normal children. The reviewer gave only the Turkish reference; you can find an English version of the article as "A mobile app that uses neurofeedback and multi-sensory learning methods improves reading abilities in dyslexia: A pilot study". Appl.Neuropsychology vol 11 2022. https://doi.org/10.1080/21622965.2021.1908897

Reviewer 1 ·

Basic reporting

I appreciate the careful responses the authors took to all reviewers. This is a much more meaningful paper.

Experimental design

None

Validity of the findings

None

Additional comments

None

Reviewer 2 ·

Basic reporting

I have read the lines between 87-112. The whole story is not like white and blacks.
Yes, for the individuals with high IQ, brain tends to spende less energy and they have less theta and delta.
But the same individuals may have higher theta and delta sometimes as a side effect due to metabolism problems, or inflamation. If the inflamation is high, this might cause their cognitive abilities to decline within a time frame. So, when you measure hight theta and lower IQ in the WISC-R tests sometimes, it does not map that this individual does not have a high IQ. These are transient, and changable states and changes over time.

About LD studies, you whould mention the following newer clinical trial as well.

EROĞLU GÜNET, TEBER SERAP, Ertürk Kardelen, Kırmızı Meltem, EKİCİ BARIŞ, ARMAN FEHİM, Balcısoy Selim, Özcan Yusuf Ziya, ÇETİN MÜJDAT (2021)
Neurofeedback ve çoklu duyusal öğrenme yöntemlerini kullanan bir mobil uygulama, dislekside okuma becerilerini geliştiriyor: Pilot çalışma. Uygulamalı Nöropsikoloji-Çocuk, Doi: 10.1080/21622965.2021.1908897 (Yayın No: 7110398)

Experimental design

In the experimental design, whether they use any drugs or take any remediation education should be stated. The control group and the experimental group numbers should be the same.

Validity of the findings

These results are partially correct.
If the underlying problem is due to the pregnancy conditions of the mother, then we may talk about the brain maturation delay.
If there would be epigenetic tendency, like Vit D receptor problems, then it also includes the metabolic and genetic variations. These epigenetic conditions are variable.

Additional comments

The article does not contribute and does not show any valuable contributions to the literature.

Reviewer 3 ·

Basic reporting

The authors replied to all my comments and did the manuscript modifications requested.
I am fully satisfied and I endorse the pubblication

Experimental design

The authors replied to all my comments

Validity of the findings

The authors replied to all my comments

Additional comments

The authors replied to all my comments

---

## Round 0.3 · accepted · Accept

Thank you for answering the reviewers' comments. This article is ready for publication.